# Systematic detection of horizontal gene transfer across genera among multidrug-resistant bacteria in a single hospital

Daniel R Evans[1,2], Marissa P Griffith[3], Alexander J Sundermann[3], Kathleen A Shutt[3], Melissa I Saul[4], Mustapha M Mustapha[3], Jane W Marsh[3], Vaughn S Cooper[5], Lee H Harrison[3], Daria Van Tyne[1]*

[1]Division of Infectious Diseases, University of Pittsburgh School of Medicine, Pittsburgh, United States; [2]Department of Infectious Diseases and Microbiology, University of Pittsburgh Graduate School of Public Health, Pittsburgh, United States; [3]Microbial Genomic Epidemiology Laboratory, Infectious Diseases Epidemiology Research Unit, University of Pittsburgh School of Medicine and Graduate School of Public Health, Pittsburgh, United States; [4]Department of Medicine, University of Pittsburgh School of Medicine, Pittsburgh, United States; [5]Department of Microbiology and Molecular Genetics, and Center for Evolutionary Biology and Medicine, University of Pittsburgh, Pittsburgh, United States

*For correspondence:
VANTYNE@pitt.edu

Competing interests: The authors declare that no competing interests exist.

**Abstract** Multidrug-resistant bacteria pose a serious health threat, especially in hospitals. Horizontal gene transfer (HGT) of mobile genetic elements (MGEs) facilitates the spread of antibiotic resistance, virulence, and environmental persistence genes between nosocomial pathogens. We screened the genomes of 2173 bacterial isolates from healthcare-associated infections from a single hospital over 18 months, and identified identical nucleotide regions in bacteria belonging to distinct genera. To further resolve these shared sequences, we performed long-read sequencing on a subset of isolates and generated highly contiguous genomes. We then tracked the appearance of ten different plasmids in all 2173 genomes, and found evidence of plasmid transfer independent from bacterial transmission. Finally, we identified two instances of likely plasmid transfer within individual patients, including one plasmid that likely transferred to a second patient. This work expands our understanding of HGT in healthcare settings, and can inform efforts to limit the spread of drug-resistant pathogens in hospitals.

## Introduction

Horizontal gene transfer (HGT) is a driving force behind the multidrug-resistance and heightened virulence of healthcare-associated bacterial infections (*Lerminiaux and Cameron, 2019*). Genes conferring antibiotic resistance, heightened virulence, and environmental persistence are often encoded on mobile genetic elements (MGEs), which can be readily shared between bacterial pathogens via HGT (*Juhas, 2015*). While rates of HGT are not well quantified in clinical settings, prior studies have shown that MGEs can mediate and/or exacerbate nosocomial outbreaks (*Bosch et al., 2017*; *Jamrozy et al., 2017*; *Martin et al., 2017*; *Sheppard et al., 2016*). Recent studies have also demonstrated that multidrug-resistant healthcare-associated bacteria share MGEs across large phylogenetic distances (*Cerqueira et al., 2017*; *Hazen et al., 2018*; *Kwong et al., 2018*). Understanding the dynamics of MGE transfer in clinical settings can uncover important epidemiologic links that are not currently identified by traditional infection control methodologies (*Lerminiaux and Cameron, 2019*; *Schmithausen et al., 2019*; *Stadler et al., 2018*).

**eLife digest** Bacteria are able to pass each other genes that make them invulnerable to antibiotics. This exchange of genetic material, also called horizontal gene transfer, can turn otherwise harmless bacteria into drug-resistant 'superbugs'. This is particularly problematic in hospitals, where bacteria use horizontal gene transfer to become resistant to several antibiotics and disinfectants at once, leading to serious infections that are difficult to treat.

How can scientists stop bacteria from sharing genes with one another? To answer this question, first it is important to understand how horizontal gene transfer happens in the bacteria that cause infections in hospitals. To this end, Evans et al. examined the genomes of over 2000 different bacteria, collected from a hospital over 18 months, for signs of horizontal transfer. First the experiments identified the genetic material that had potentially been transferred between bacteria, also known as 'mobile genetic elements'. Next, Evans et al. examined the data of patients who had been infected with the bacteria carrying these mobile genetic elements to see whether horizontal transfer might have happened in the hospital.

By combining genomics with patient data, it was determined that many of the mobile genetic elements identified were likely being shared among hospital bacteria. One of the mobile genetic elements identified was able to provide resistance to several drugs, and appeared to have been horizontally transferred between bacteria infecting two separate patients.

The findings of Evans et al. show that the horizontal transfer of mobile genetic elements in hospital settings is likely frequent, but complex and difficult to study with current methods. The results of this study show how these events can now be tracked and analyzed, which may lead to new strategies for controlling the spread of antibiotic resistance.

Methods to identify and track the movement of MGEs among bacterial populations on short timescales are limited. Bacterial whole-genome sequencing has transformed infectious disease epidemiology over the last decade (*Ladner et al., 2019*), providing powerful new tools to identify and intervene against outbreaks (*Sundermann et al., 2019b*). Despite these advances, efforts to track MGE movement have focused almost exclusively on drug resistance and virulence genes (*Cerqueira et al., 2017*; *Hardiman et al., 2016*; *Martin et al., 2017*; *Stadler et al., 2018*), often ignoring the broader genomic context of the mobile elements themselves. Many studies rely on the identification of plasmid replicons, transposases, and other 'marker genes' (*Orlek et al., 2017*), an approach that oversimplifies the diversity of MGEs and may lead to incomplete or erroneous conclusions about their epidemiology. While querying databases containing curated MGE-associated sequences is useful for the rapid screening of clinical isolates for known MGEs, it will not capture novel MGEs. Additionally, whole-genome sequencing using short-read technologies generates genome assemblies that usually do not resolve MGE sequences, due to the abundance of repetitive elements that MGEs often contain (*Arredondo-Alonso et al., 2017*). Advances in long-read sequencing can mitigate this problem; hybrid assembly of short- and long-read sequence data allows the genomic context of chromosomal and extrachromosomal MGEs to be precisely visualized (*Cerqueira et al., 2017*; *Conlan et al., 2014*; *George et al., 2017*). Finally, studying the epidemiology of MGEs in clinical settings requires detailed individual-level patient clinical data, without which HGT occurrence in the hospital cannot be identified (*Conlan et al., 2014*).

Here, we performed an alignment-based screen for shared nucleotide sequences in a large and diverse collection of bacterial genomes sampled from infections within a single hospital over an 18-month time period. With this approach, we identified shared sequences that occurred in the genomes of bacteria belonging to different genera. Because they were identical, we suspect that these sequences recently transferred between bacteria within the hospital setting. Further analysis using long-read sequencing and reference-based resolution of distinct MGEs enabled us to precisely characterize MGE architecture and cargo, and to track MGE occurrence over time. Cross-referencing our results with available patient metadata allowed us to follow these elements as they emerged and were maintained among nosocomial bacterial populations.

## Results

### Identification of nucleotide sequences shared across bacterial genera in a single hospital

Our experimental workflow is depicted in *Figure 1A*. To identify genetic material shared between distantly related bacteria in the hospital setting, we screened a dataset containing 2173 whole-genome sequences of clinical isolates of high-priority Gram-positive and Gram-negative bacteria collected from a single hospital over an 18-month period beginning in November 2016 as part of the Enhanced Detection System for Hospital-Acquired Transmission (EDS-HAT) project at the University of Pittsburgh (*Sundermann et al., 2019a*) (Methods and *Supplementary file 1*). To have maximal contrast, we focused on identical sequences found in the genomes of bacteria belonging to different genera. We performed an all-by-all alignment of the 2173 genomes in the dataset using nucmer (*Marçais et al., 2018*), and filtered the results to retain alignments of at least 5 kb that shared 100% identity between bacteria of different genera. The resulting sequences were extracted and clustered using Cytoscape (*Figure 1B*). We also explored alignments > 3 kb and >10 kb, and found that the number of clusters identified was highly dependent upon the alignment length cut-off used (*Figure 1—figure supplement 1*). We chose to use 5 kb for our analysis because of the intermediate number of resulting clusters. This approach identified shared sequences in 196 genomes belonging to 11 genera, which were grouped into 51 clusters of related sequences (*Supplementary file 2*). We compared the patient demographics and clinical features of the subset of patients from whom the 196 isolates encoding shared sequence clusters were derived with the other patients in the dataset (*Table 1*). While patient demographics were similar between groups, isolates encoding shared sequence clusters were cultured from patients with more co-morbidities (as measured by Charlson co-morbidity index, p=0.03), and with higher rates of solid organ transplant (p=0.02) (*Table 1*).

The shared sequence clusters we identified ranged in size from two to 52 genomes and comprised two, three, or four different genera (*Figure 1B*). Shared sequences were found predominantly among Gram-negative *Enterobacteriaceae*, particularly *Klebsiella spp.*, *Escherichia coli*, and *Citrobacter spp.* (*Figure 1C*). Annotation of clustered sequences confirmed that more than 80% of clusters encoded one or more genes involved in DNA mobilization, such as plasmid replication, integration, or other mobile functions presumably involved in HGT (*Figure 1D* and *Supplementary file 2*). Approximately one-quarter of the clusters contained antimicrobial resistance genes, including genes encoding resistance to aminoglycosides, antifolates, beta-lactams, macrolides, quinolones, sulphonamides, and tetracyclines (*Figure 1D and E*). Finally, 8 of 51 clusters encoded genes and operons whose products were predicted to interact with metals, including arsenic, copper, mercury, nickel, and silver (*Figure 1D*). Collectively, these results indicate that our systematic, alignment-based method successfully identified sequences associated with MGEs, particularly in pathogens known to engage in HGT (*Huddleston, 2014*; *Juhas, 2015*).

To assess the phylogenetic distribution of the shared sequence clusters we identified, we constructed a core gene-based phylogeny of the 196 genomes encoding one or more clusters using the Genome Taxonomy Database Tool Kit (GTDBTK) (*Parks et al., 2018*; *Figure 2*). Shared sequence clusters were often found among bacteria in related genera, in particular the *Enterobacteriaceae*. We did not observe any shared sequences that were present in both Gram-positive and Gram-negative isolate genomes, but we did find shared sequences in the genomes of distantly related bacteria. For example, we identified a shared sequence cluster comprised of three aminoglycoside resistance genes that was identical between a vancomycin resistance-encoding plasmid carried by *Enterococcus faecium* and the *Clostridioides difficile* chromosome (cluster C9, *Figure 3A*). The *C. difficile* strain carrying this element was previously found to also harbor an *npmA* aminoglycoside resistance gene (*Marsh et al., 2019b*). Separately, we found a section of an integrative conjugative element that was identical between two *Pseudomonas aeruginosa* isolates and one *Serratia marcescens* isolate (cluster C30, *Figure 3B*). Identical regions of this element included formaldehyde resistance genes and Uvr endonucleases. Finally, we detected complete and identical Tn7 transposons in the genomes of *Acinetobacter baumannii*, *E. coli*, and *Proteus mirabilis* isolates (cluster C17, *Figure 3C*). The Tn7 sequence we detected was also identical to the Tn7 sequence of pR721, an *E. coli* plasmid that was first described in 1990 and was sequenced in 2014 (*Komano et al., 1990*).

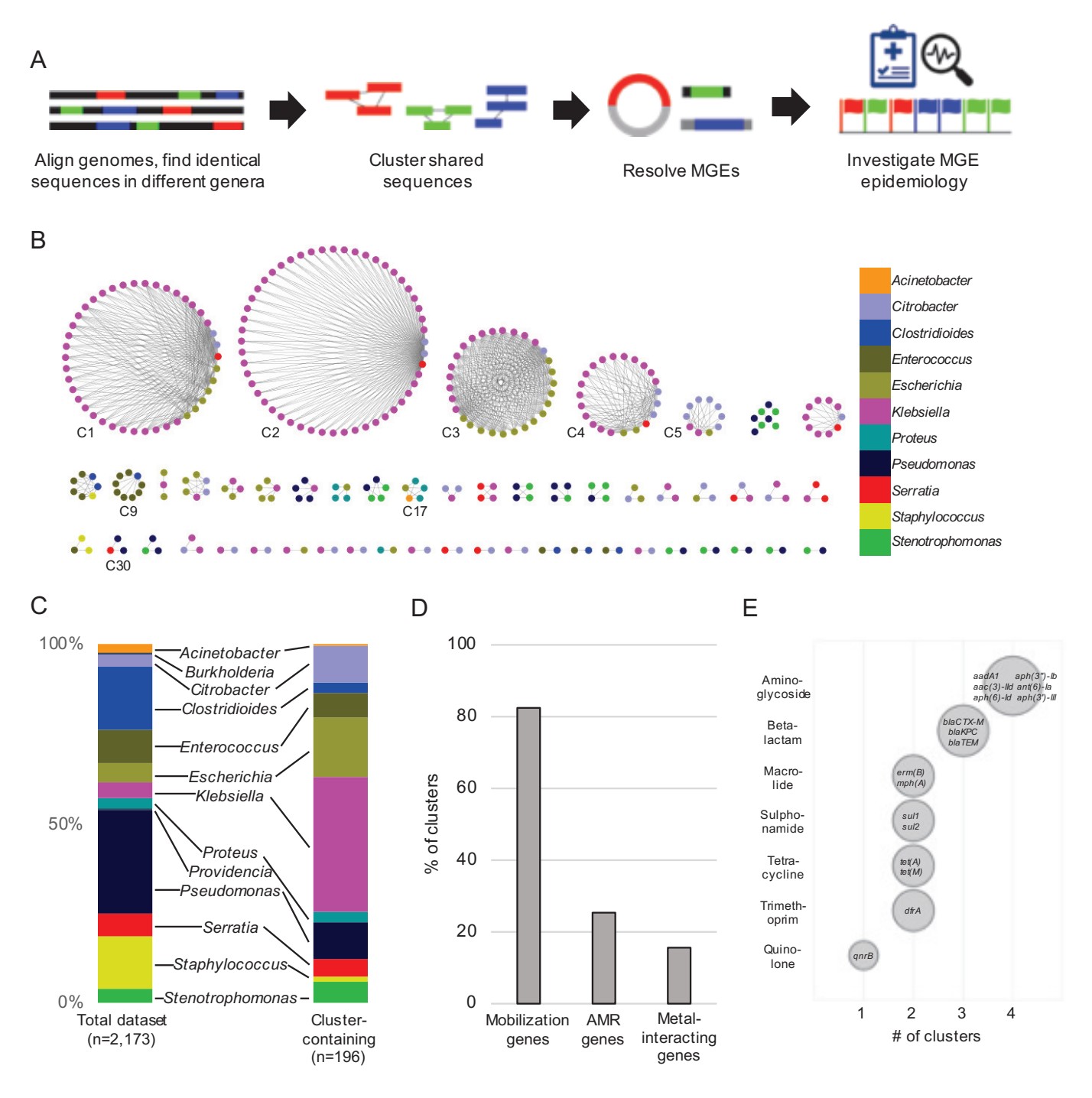

**Figure 1.** Identification of nucleotide sequences shared across bacterial genera in a single hospital. (**A**) Approach to identify shared sequence clusters, and then resolve the MGEs that carry them. (**B**) 51 clusters of shared sequences found in distinct genera visualized with Cytoscape. Nodes represent bacterial isolates and are color-coded by genus. Edges connect nodes from different genera sharing >5 kb of sequence at 100% nucleotide sequence identity. Clusters examined more closely in subsequent figures are labeled. (**C**) Genus distribution of all 2173 genomes in the dataset (left) and the 196 isolates encoding one or more shared sequence clusters (right). (**D**) Prevalence of mobilization, antimicrobial resistance (AMR) and metal-interacting genes among 51 shared sequence clusters. (**E**) Summary of AMR genes identified in shared sequence clusters. Genes are grouped by antibiotic class, and bubble sizes correspond to prevalence among the clusters shown in (**B**). AMR gene names are listed inside each bubble. To generate (**D**) and (**E**) the longest sequence in each cluster was examined.

The online version of this article includes the following figure supplement(s) for figure 1:

*Figure 1 continued on next page*

*Figure 1 continued*

**Figure supplement 1.** Shared sequence clustering based on alternate sequence length parameters.

## Shared sequences often reside on MGEs in different combinations and contexts

To further investigate the genomic context of the shared sequence clusters we identified, we selected the isolate containing the longest sequence in each cluster from C1-C5 for long-read sequencing using Oxford Nanopore technology. Hybrid assembly combining short Illumina reads and long Nanopore reads generated highly contiguous chromosomal and plasmid sequences, which allowed us to resolve MGEs carrying one or more of the most prevalent shared sequence clusters (*Table 2*). We found that several of the shorter and more prevalent shared sequences were carried on a variety of different plasmid and chromosomal MGEs, and furthermore, the sequences co-occurred in different orders, orientations, and combinations (*Table 2*, *Figure 4A*). This kind of 'nesting' of mobilizable sequences within larger MGEs has been previously observed (*Sheppard et al., 2016*), and our findings further support the mosaic, mix-and-match nature of the shorter shared sequences we identified. We also confirmed that these shared sequences were indeed mobilizable, since they were found independently within multiple distinct, larger MGEs. A closer examination of the three largest shared sequence clusters (C1, C2, C3) showed that C1 sequences did not all share a common 'core' nucleotide sequence, but rather could be aligned in a pairwise fashion to generate a contiguous 'chain' of sequences (*Figure 4B*). Clusters C2 and C3, on the other hand, did contain 'core' sequences that were present in all genomes containing the cluster (*Figure 4C and D*).

## Plasmids carrying shared sequence clusters are found in bacteria belonging to multiple sequence types, species, and genera circulating in the same hospital

More than half (104/196) of the genomes encoding shared sequence clusters contained one or more of the five most prevalent clusters (C1-C5, *Figure 1B*). In all five cases, the shared sequences were short (usually less than 10 kb), and they were predicted to be carried on plasmids shared between *Enterobacteriaceae*. We set out to resolve the genomic context of each of these five clusters in all isolates containing them. We used an iterative approach that started with long-read sequencing and hybrid assembly of the earliest isolate in each cluster to generate reference sequences of cluster-containing MGEs (chromosomal or plasmid) (*Supplementary file 3*). Then we mapped contigs from Illumina-only assemblies to the MGE reference sequences to assess their coverage in other genomes, using a cutoff of >90% coverage to define an MGE as potentially transferred between

**Table 1.** Demographics and co-morbidities of study patients.

| | All isolates | Shared sequence isolates | p-value[†] |
|---|---|---|---|
| Total number of isolates | 2173 | 196 | |
| Number of unique patients | 1533 | 172 | |
| **Demographics[*]:** | n = 1445 | n = 157 | |
| Median age, years (range) | 62 (17–98) | 63 (19–89) | 0.89 |
| Male gender | 738 (51%) | 81 (52%) | 0.93 |
| **Co-morbidities:** | | | |
| Median Charlson Co-morbidity Index (range) | 3 (0–15) | 4 (0–13) | **0.03** |
| Solid organ transplant | 180 (12%) | 29 (18%) | **0.02** |
| Diabetes mellitus | 369 (26%) | 42 (27%) | 0.7 |
| Cystic fibrosis | 31 (2%) | 5 (3%) | 0.37 |

[*]Demographics and co-morbidities are reported for patients for whom information was available.

[†]p-values were calculated using Fisher's Exact test for categorical variables and Wilcoxon rank-sum test for continuous variables. Shared sequence isolates were removed from the 'all isolates' group to assess the significance of differences between groups.

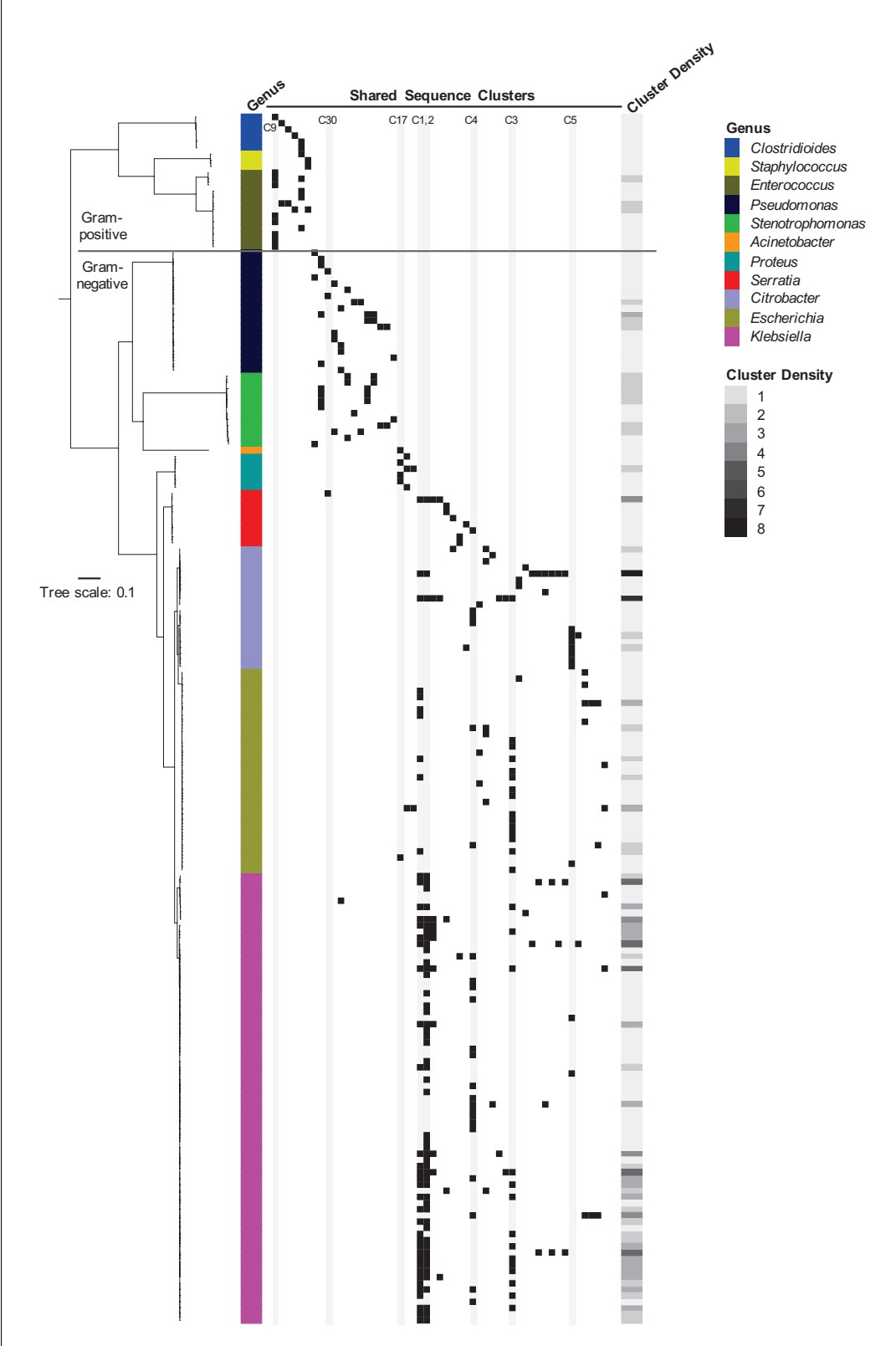

**Figure 2.** Phylogenetic distribution of shared sequence clusters across 196 genomes. A phylogeny was made by aligning amino acid sequences of 120 ubiquitous protein coding genes from the Genome Taxonomy Database Tool Kit. The scale bar shows the number of amino acid substitutions per site. Black squares mark the presence of one or more clusters in each genome, with each column corresponding to a different cluster. The heat map to the

*Figure 2 continued on next page*

*Figure 2 continued*

right shows cluster density (i.e. total number of cross-genus shared sequence clusters) in each bacterial genome. Clusters examined more closely in subsequent figures are labeled and shaded in gray.

isolates (Materials and methods). This approach allowed us to query the presence of MGEs from genomes sequenced with Illumina technology alone, without requiring long-read sequencing of all isolates or relying on external references. We found that 11 of the 104 isolates (all *E. coli*) carried cluster C1 and C3 sequences on their chromosome, while the remaining 93 isolates carried cluster C1-C5 sequences on 17 distinct plasmids. Seven of these plasmids were present in only one isolate in the dataset, but ten plasmids appeared to be shared between more than one isolate (*Table 2*, *Figure 5*). We also conducted the same reference-based coverage analysis for all 2173 genomes in the original dataset, and identified an additional 16 isolates with >90% coverage of an MGE encoding C1-C5 sequences (*Supplementary file 4*).

While all the shared sequences we originally identified were present in the genomes of bacteria belonging to different genera, the plasmids that we resolved were variable in how widely they were shared. For example, two plasmids were only found among isolates belonging to a single species and multilocus sequence type (ST), suggesting that they were likely transmitted between patients along with the bacteria that were carrying them (*Figure 5A*). These included an IncF *blaKPC-3* carbapenemase-encoding plasmid (pKLP00149_2) found in 17 *K. pneumoniae* isolates belonging to ST258, a multidrug-resistant and highly virulent hospital-adapted bacterial lineage that has recently undergone clonal expansion in our hospital (*Marsh et al., 2019a*). All isolates carrying this plasmid belonged to Clade II of ST258, which has caused multiple outbreaks at our center (*Figure 5—figure supplement 1*; *Marsh et al., 2019a*). We also found an IncF *blaOXA-1* extended spectrum beta-lactamase-encoding plasmid in eight *E. coli* isolates belonging to ST131, another multidrug-resistant and hypervirulent clone (*Manges et al., 2019*). As above, this plasmid was found in closely related

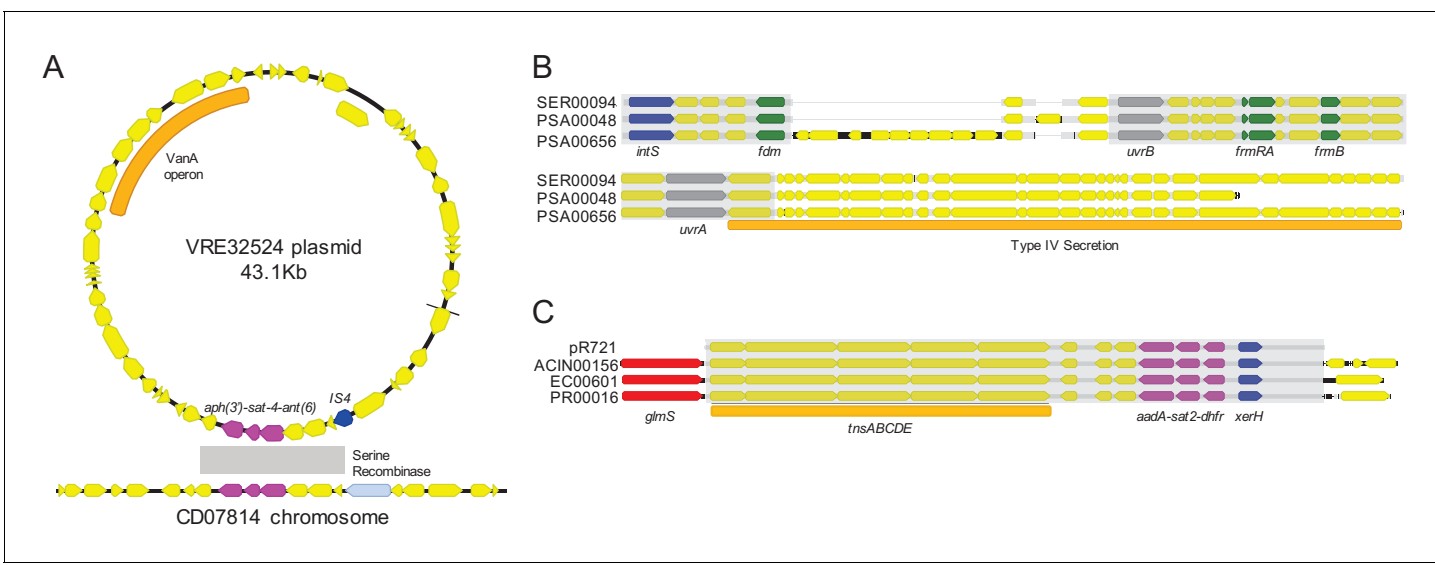

**Figure 3.** Examples of sequence sharing across genera. (**A**) Genes shared between a vancomycin-resistant *E. faecium* (VRE) plasmid and a *C. difficile* chromosome (cluster C9). The VanA operon, conferring vancomycin resistance, is marked with an orange bar. Shared drug resistance genes are colored magenta, and mobilization genes are colored blue. Gray shading marks DNA sequence that is 100% identical between isolates. (**B**) Identical portions of an integrated conjugative element (cluster C30) shared between an *S. marcescens* genome (SER00094) and two *P. aeruginosa* genomes (PSA00048 and PSA00656). Blue = *intS* integrase; green = formaldehyde resistance genes; gray = UvrABC system genes. Type IV secretion machinery is marked with an orange bar, and gray shading marks sequences that are 100% identical between isolates. (**C**) Identical Tn7 transposons shared between *A. baumannii*, *E. coli*, and *P. mirabilis* (cluster C17). The Tn7 sequence of the pR721 plasmid is shown at the top. The *tnsABCDE* transposon machinery is marked with an orange bar, and the *glmS* gene, which flanks the Tn7 insertion site, is colored red. Shared drug resistance genes are colored magenta, and an *xerH* tyrosine recombinase is colored blue. Gray shading marks sequences that are 100% identical between isolates.

**Table 2.** Resolved MGEs and associated antibiotic resistance and metal interaction gene contents.

| MGE[*] | Length (kb) | % GC | Replicons[†] | MOB Family[‡] | Antibiotic resistance Genes[§] | Metal interaction Genes[¶] |
|---|---|---|---|---|---|---|
| cEC00609 | 39.1 | 52.6 | None | None | *aac(3)-IIa, aac(6')-Ib-cr, blaCTX-M-1, blaOXA-1, catB3, tet(A)* | None |
| pCB00017_2 | 196.8 | 51.7 | FIB, FIIK | MOB-F | *aac(6')-Ib-cr, aph(3'')-Ib, aph(6)-Id, blaCTX-M-15, blaOXA-1, blaTEM-1B, catB3, qnrB1, tet(A), sul2* | *copD* operon, *pcoE, silE, silP, ars* operon |
| pCB00028_2 | 383.1 | 47.5 | HI2, HI2A | MOB-H | *aac(3)-IIa, aac(6')-Ib-cr, aadA1, aph(3'')-Ib, aph(6)-Id, blaCTX-M-15, blaOXA-1, baTEM-1B, catA1, catB3, dfrA14, sul2, tet(A)* | *pcoE, merR, merB* |
| pEC00668_2 | 145.4 | 55.9 | FIA, FII | MOB-F | *aac(6)-Id, aph(3'')-Ib, dfrA14, blaTEM-1B, mph(A), sul2* | *efeU, merA, merC, merP, merR, merT* |
| pEC00690_2 | 106.8 | 54.7 | FIA, FII | MOB-F | *aac(6')-Ibcr, blaOXA-1, catB3, tet(A)* | *efeU* |
| pKLP00149_2 | 165.2 | 52.9 | FIIY | MOB-F | *aac(6')-Ib, aac(6')-Ib-cr, aadA1, aph(3'')-Ib, aph(6)-Id, blaKPC-3, blaOXA-9, blaSHV-182, blaTEM-1A, dfrA14, sul2* | *csoR* |
| pKLP00155_6 | 9.5 | 54.9 | ColRNAI | MOB-C | None | None |
| pKLP00161_2 | 236.5 | 55.1 | FIB, FIIK | MOB-F | *aac(6')-Ib-cr, aph(3'')-Ib, aph(6)-Id, blaCTX-M-15, blaOXA-1, blaTEM-1B, dfrA14, qnrB1, sul2, tet(A)* | *copD* operon, *pcoC, pcoE, silE, silP, ars* operon |
| pKLP00177_3 | 170.8 | 52.0 | FIB | MOB-F | *aac(3)-IIa, aac(6')-Ib-cr, aph(3'')-Ib, aph(6)-Id, blaCTX-M-15, blaOXA-1, blaTEM-1B, catB3, dfrA14, qnrB1, sul2, tet(A)* | *copD* operon, *pcoC, pcoE, silE, silP, ars* operon |
| pKLP00182_3 | 15.8 | 51.2 | A/C | MOB-H | *aac(6')-Ib-cr, blaOXA-1, catB3, dfrA14, tet(A)* | None |
| pKLP00215_4 | 113.6 | 53.9 | FIB, FIIK | MOB-F | *blaKPC-2, blaOXA-9, blaTEM-1A* | *merB, merR* |
| pKLP00218_2 | 164.7 | 54.9 | FIB, FIIK | MOB-F | *aph(3'')-Ib, aph(6)-Id, blaCTX-M-15, blaTEM-1B, dfrA14, sul2* | *copD* operon, *pcoC, pcoE, silE, silP, ars* operon |
| pKLP00221_2 | 242.3 | 53.2 | ColRNAI, FIB, FII | MOB-C, MOB-F | *aac(6')-Ib, aada2, aph(3')—1a, blaKPC-2, blaOXA-9, blaTEM-1A, catA1, dfrA12, mph(A), sul1* | *copD* operon, *pcoC, pcoE, silE, silP, ars* operon |

[*]MGE names include location (c = chromosome, p=plasmid), name of the reference isolate sequenced, and assembly contig number (_2, _3, _4, _6).

[†]Replicons were identified by querying Plasmid MLST and PlasmidFinder databases.

[‡]MOB families were assigned with MOBscan.

[§]Antibiotic resistance genes were identified by querying the ResFinder database.

[¶]Metal interaction genes were identified by examining annotations assigned by Prokka.

ST131 isolates (*Figure 5—figure supplement 1*), suggesting that it was vertically transmitted along with the bacteria carrying it.

In addition to plasmids that occurred in bacteria belonging to the same ST, we also identified plasmids that were present in isolates belonging to different STs of the same species, or in different species of the same genus (*Figure 5B*). All isolates in this case were *K. pneumoniae* or *K. oxytoca*, suggesting widespread sharing of plasmids between distinct *Klebsiella* species and STs. The plasmids all carried antibiotic resistance genes, and many also carried metal interaction genes (*Table 2*). Finally, we identified three different plasmids that were shared between different bacterial genera all belonging to the *Enterobacteriaceae* (*Figure 5C*). One 9.5 kb ColRNAI plasmid (pKLP00155_6) carrying the colicin bacterial toxin was found in 26 isolates belonging to 10 different STs and 4 different genera. Taken together, these results indicate that some plasmids carrying putative MGEs were likely inherited vertically as bacteria were transmitted between patients in the hospital, while others appear to have transferred independently of bacterial transmission.

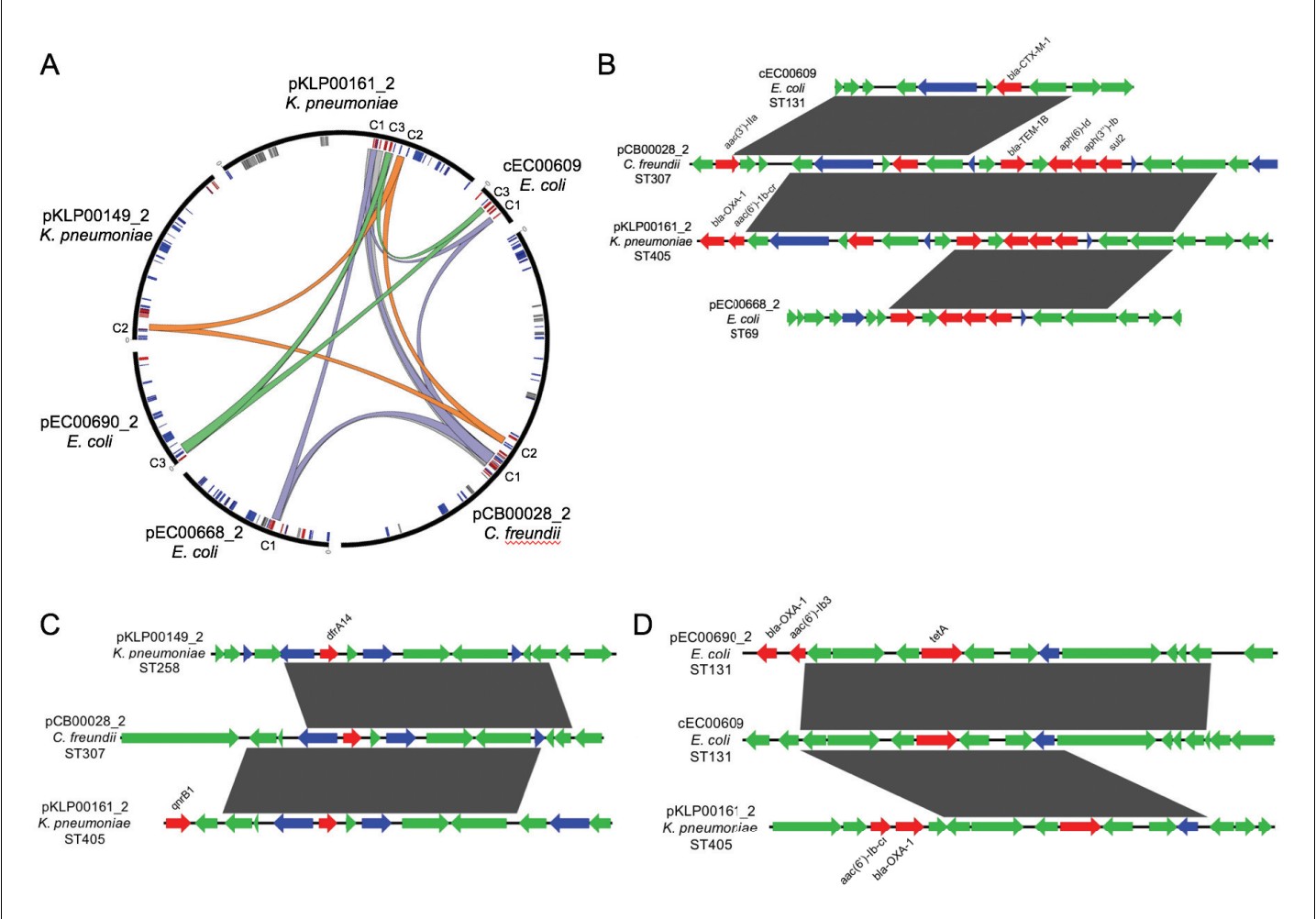

**Figure 4.** Mosaicism of shared sequence clusters present on diverse MGEs. (**A**) Circos plot of six distinct MGEs (black bars) that encode shared sequence clusters C1, C2, and C3. Lowercase letters in sequence names indicate element type (c = chromosome, p=plasmid). Homologous cluster sequences are connected to one another with colored links (purple = C1, orange = C2, green = C3, gray = other). Inner circle depicts genes involved in mobilization (blue), antibiotic resistance (red) and metal interaction (gray). (**B–D**) Alignments of sequences grouped into shared sequence clusters C1 (**B**), C2 (**C**), and C3 (**D**) from the MGEs displayed in (**A**). ORFs are colored by function (blue = mobilization, red = antibiotic resistance, green = other/ hypothetical). Antibiotic resistance genes are labeled above and dark gray blocks connect sequences that are identical over at least 5 kb.

## Likely HGT across genera within individual patients

By cross-referencing the isolates containing shared plasmids with de-identified patient data, we found two instances of identical plasmids present in pairs of isolates of different genera that were collected from the same patient, on the same date, and from the same sample source (*Figure 6*). A *K. pneumoniae* ST405 isolate (KLP00215) and an *E. coli* ST69 isolate (EC00678) collected from a tissue infection from Patient A each harbored a 113.6 kb IncF plasmid carrying *blaKPC-2*, *blaOXA-9*, and *blaTEM-1A* enzymes, as well as a mercury detoxification operon (*Figure 6A,B*). An isolate from a second patient (Patient B, EC00701, *E. coli* ST131), which was cultured 109 days after the isolates from Patient A, also encoded a nearly identical plasmid. A systematic chart review for Patients A and B revealed that they occupied adjacent hospital rooms for four days during a time period after Patient A's isolates were collected but before Patient B's isolate was collected. During this time, the two patients were treated by the same healthcare staff, who might have transferred bacteria between them.

In the second case of putative within-patient HGT, a *K. pneumoniae* ST231 isolate (KLP00187) and a *Citrobacter braakii* ST356 isolate (CB00017) were both collected from the same urine sample of Patient C (*Figure 6C*). Both isolates carried nearly identical 196.8 kb IncF plasmids conferring

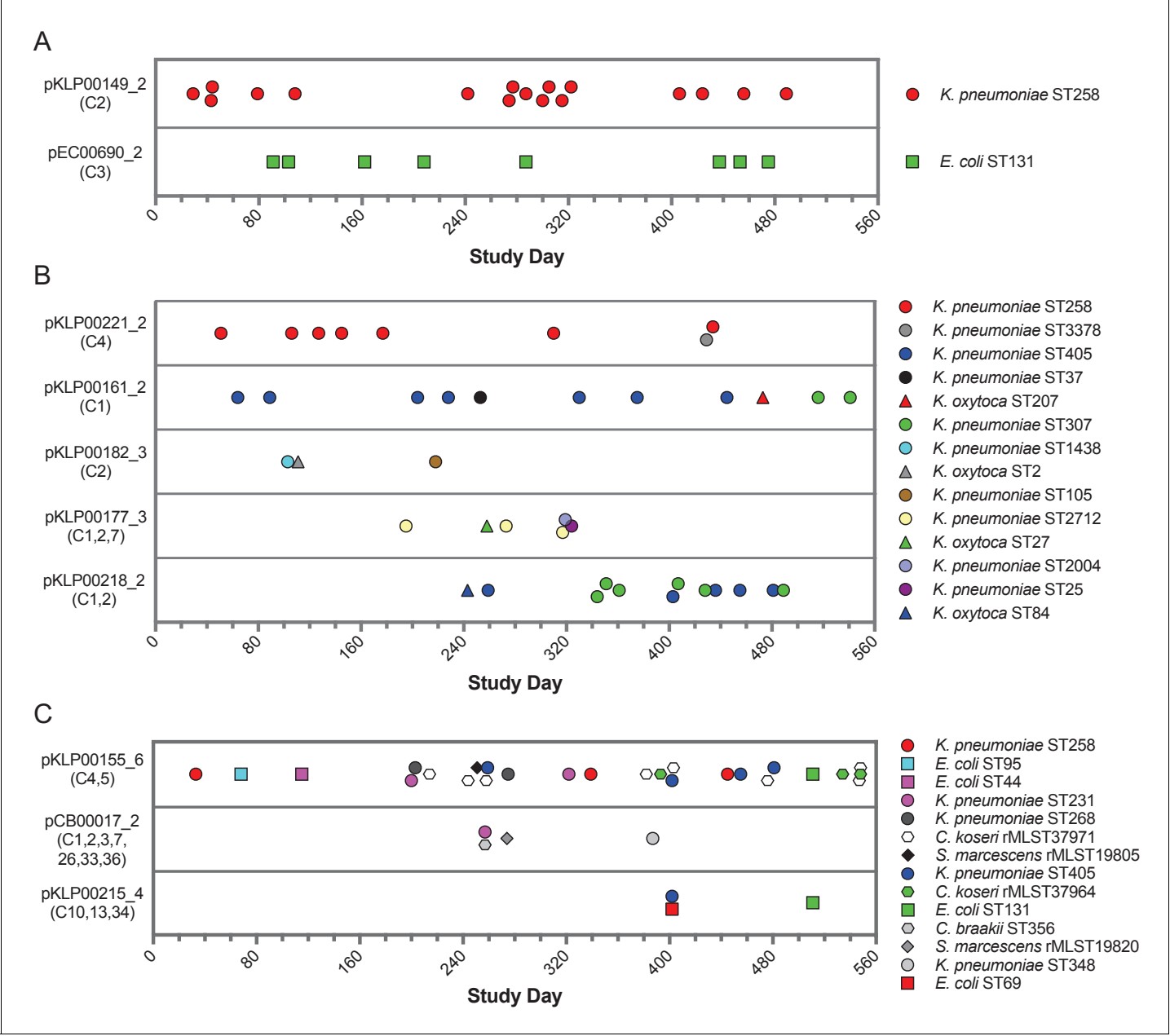

**Figure 5.** Timelines of plasmid occurrence among isolates of the same ST (**A**), same genus (**B**), or different genera (**C**). Illumina contigs of all study isolate genomes were mapped to the reference plasmid sequences indicated to the left of each panel, and plasmids were called 'present' if an isolate genome of any genus contained >90% of the reference sequence (based on mapping coverage). Timelines show the study date of each isolate, and the shared sequence clusters carried by each plasmid are listed in parentheses below the plasmid names. Shape and color of data points correspond to bacterial species and ST, respectively. More information about each plasmid is provided in *Table 2*.

The online version of this article includes the following figure supplement(s) for figure 5:

**Figure supplement 1.** Phylogenetic context of hospital-associated infection isolates carrying the same plasmids.

resistance to aminoglycosides, beta-lactams, chloramphenicol, fluoroquinolones, sulfonamides, tetra-cyclines, and trimethoprim, as well as operons encoding copper and arsenic resistance (*Table 2*). Furthermore, isolates from two subsequent patients (Patient D and Patient E) also carried plasmids that were similar to the plasmid shared between KLP00187 and CB00017. Alignment of the sequences of all four plasmids showed that the plasmids isolated from Patient C were nearly identical, while the plasmids from Patients D and E had small differences in their gene content and organization

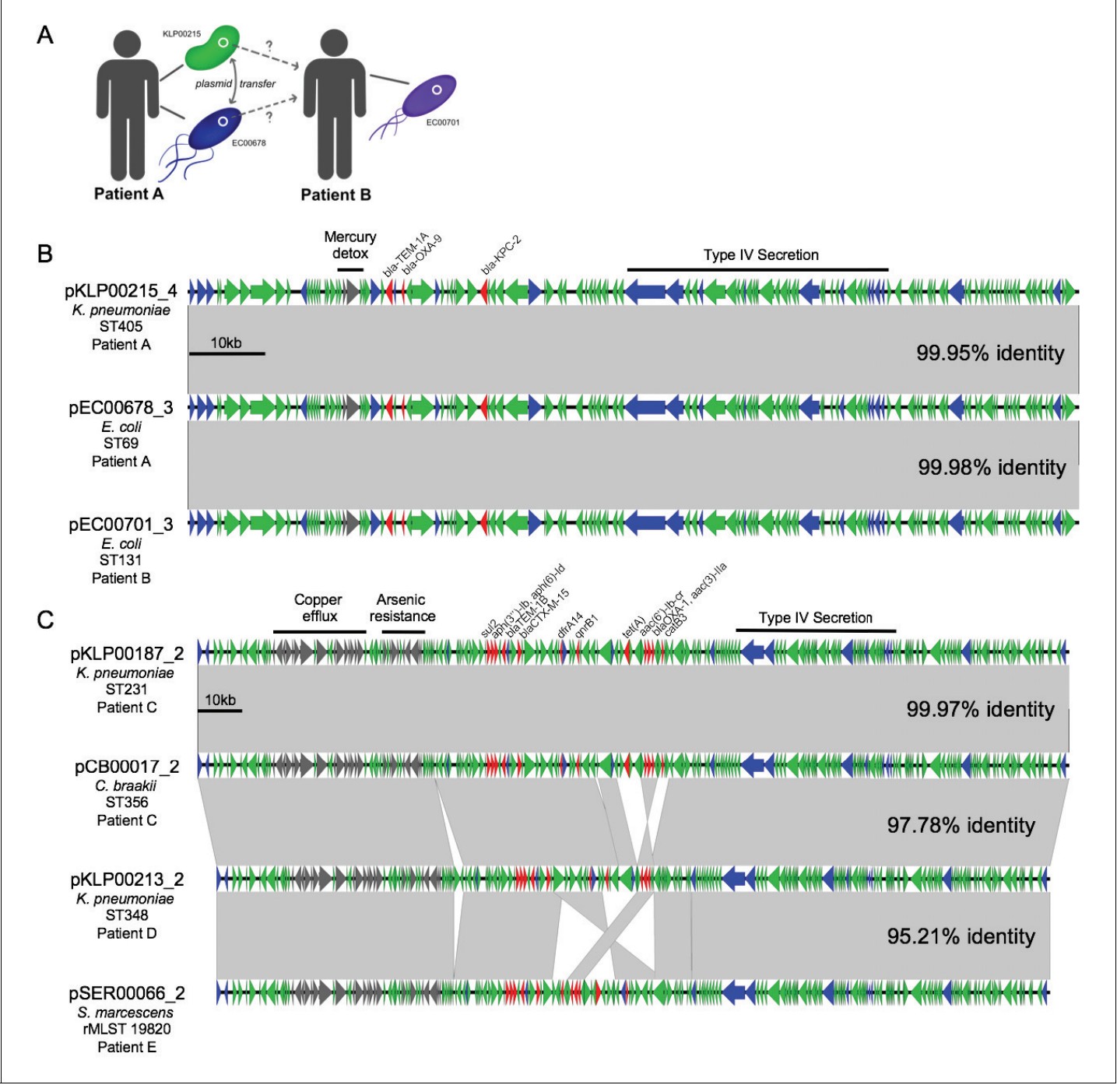

**Figure 6.** Cross-genus transfer of plasmids within and between patients. (**A**) Schematic diagram showing *K. pneumoniae* and *E. coli* isolates bearing the same plasmid sampled from two patients. (**B**) Nucleotide alignment of the plasmid presumably transferred within and between the patients shown in (**A**). A 113.6 kb IncF carbapenemase-encoding plasmid was resolved from two genomes of different bacterial isolates from the same clinical specimen from Patient A. A nearly identical plasmid was also identified in an isolate from Patient B, who occupied a hospital room adjacent to Patient A. (**C**) Alignment of a 196.8 kb IncF multidrug-resistance plasmid resolved from two genomes of different bacterial isolates from the same clinical specimen from Patient C. Similar plasmids were also found in isolates from two additional patients (Patient D and Patient E), who had no identifiable epidemiologic links with Patient C. ORFs are colored by function (blue = mobilization, red = antibiotic resistance, gray = metal interacting, green = other/hypothetical). Antibiotic resistance genes, metal-interacting operons, and Type IV secretion components are labeled. Gray blocks between sequences indicates regions > 5 kb with >99.9% identity, and pairwise identities across the entire plasmid are noted to the right.

(*Figure 6C*). A systematic chart review did not identify any strong epidemiologic links between the three patients, suggesting that this plasmid was not passed directly between these patients and might instead have transferred via additional bacterial populations that were not sampled.

## Discussion

Through this study, we have produced a high-resolution view of shared sequence and MGE dynamics among clinical bacterial isolates collected over an 18-month period from a single hospital. We identified, clustered, and characterized identical sequences found in multiple distinct genera, and in the process uncovered both expected and unexpected cases of shared sequence occurrence. We confirmed that some of the most common shared sequences identified were fragments of larger MGEs. We performed long-read sequencing to resolve these larger elements, and in doing so we characterized a large diversity of drug resistance-encoding plasmids. When we traced the presence of various plasmids over time, we found some that were likely transferred vertically along with the bacteria carrying them, and others that appeared to be transferred horizontally between unrelated bacteria.

Our study adds to the body of knowledge of HGT in hospital settings in new and important ways. We analyzed a large set of clinical isolates collected from a single health system, and used a systematic approach to identify shared nucleotide sequences, regardless of their type or gene content. While prior studies have used genomic epidemiology to study how HGT contributes to the transmission, persistence, and virulence of bacterial pathogens (*Bosch et al., 2017*; *Martin et al., 2017*; *Schweizer et al., 2019*; *Valenzuela et al., 2007*), the technical challenges of resolving MGEs from whole-genome sequencing data have limited the scope of these findings (*Arredondo-Alonso et al., 2017*). Furthermore, while rates of HGT between pathogenic bacteria have been quantified in vitro, very little information is currently available to assess rates of HGT in vivo or in clinical settings (*Leclerc et al., 2019*). Other studies have deliberately tracked HGT in healthcare settings by focusing either on mobile genes of interest, such as those encoding drug resistance (*Cerqueira et al., 2017*; *Hardiman et al., 2016*; *Hazen et al., 2018*), or on specific classes of MGEs (*Savinova et al., 2019*). Both of these approaches likely generate incomplete accounts of the extent of HGT in clinical settings. For this reason, we selected a pairwise alignment-based approach, whereby we only looked for identical sequences in the genomes of very distantly related bacteria. In doing so, we did not limit ourselves to only focusing on 'known' MGEs, and thus obtained a more accurate and comprehensive overview of the dynamics of HGT between bacterial genera in our hospital.

What might cause horizontally-transferred nucleotide sequences to be found at very high identity within phylogenetically distinct bacteria? Among many possible causes, we could consider the following: (1) the sequences we identified could have been recently transferred and not have had time to diverge from one another; (2) they could already be well adapted to optimally perform their functions; or (3) they could represent genetic elements that are highly intolerant to mutation. We suspect that our dataset contains all three cases. First, in the instances of likely within-patient HGT, both plasmids isolated from the same patient were nearly identical to one another. This suggests that if mutation rates of plasmids are similar to bacterial chromosomes, these plasmids would have transferred shortly before the bacteria were isolated. In both cases of likely within-patient HGT we also observed similar plasmids in the genomes of isolates from other patients, but we identified a likely route of transfer between patients only in the case where the subsequent plasmid was also nearly identical. This finding supports our theory that high plasmid identity is evidence of recent transfer. Second, the plasmids that we identified only in ST258 *K. pneumoniae* or in ST131 *E. coli* are likely well adapted to these lineages, perhaps because plasmid-imposed fitness costs have already been resolved through compensatory adaptations (*San Millan, 2018*). Third, the Tn7 transposon sequence we uncovered, which was identical in bacterial isolates from three different genera, was also identical to over two dozen publicly available genome sequences queried through a standard NCBI BLAST search. The insertion of the Tn7 transposon downstream of *glmS* in all of our isolates suggests TnsD-mediated transposition (*Parks and Peters, 2009*, p. 7), but the reason why the entire transposon sequence remains so highly conserved remains unclear.

The vast majority of shared sequences identified through our approach contained signatures of mobile elements, and our follow-up work demonstrated that these sequences could very likely move independently and assemble in a mosaic fashion on larger mobile elements like plasmids, integrative

conjugative elements, and other genomic islands. Antibiotic resistance genes were present in only a subset of the shared sequence clusters we identified, which was somewhat surprising given how many resistance genes are known to be MGE-associated. Our follow-up analysis showed, however, that resistance genes were indeed highly prevalent among many of the MGEs that we resolved. This finding is consistent with a recent study of clinical *K. pneumoniae* genomes, which showed that while antibiotic resistance genes were largely maintained at the population level, they were variably present on different MGEs that fluctuated in their prevalence over time (*Ellington et al., 2019*). Finally, we were surprised by the large number of metal-interacting genes and operons within the shared sequences that we identified. While metal-interacting genes and operons have been hypothesized to confer disinfectant tolerance and increased virulence (*Chandrangsu et al., 2017*; *McDonnell and Russell, 1999*), precisely how these elements might increase bacterial survival in the hospital environment and/or contribute to infection requires further study.

Identification of risk factors and common exposures for HGT has previously been proposed (*Conlan et al., 2014*; *Hardiman et al., 2016*; *Lerminiaux and Cameron, 2019*; *Pecora et al., 2015*), but the results of prior efforts have been limited because large genomic datasets from single health systems with corresponding epidemiologic data have not been widely available (*Struelens, 1998*). The use of routine whole-genome sequencing for outbreak surveillance in our hospital has allowed us to begin to study how the horizontal transfer of MGEs might be similar or different from bacterial transmission. In addition to finding evidence of vertical transfer of plasmids accompanying bacterial transmission, we also identified several cases in which the same MGE was identified in two or more isolates of different sequence types, species, or genera. In some cases, these isolates were collected within days or weeks of one another. This finding highlights the frequent movement of MGEs between bacterial populations, particularly in hospitalized patients (*Huddleston, 2014*; *Lerminiaux and Cameron, 2019*), and points to the importance of pairing genome sequencing with epidemiologic data to uncover routes of MGE transmission.

There are several limitations to our study. First, the dataset that we used only contained genomes of isolates from clinical infections from a pre-selected list of species, and did not include environmental samples or isolates from patient colonization. In the case of between-patient plasmid transfer that we identified, we do not know exactly how the plasmid was transferred from Patient A to Patient B because we did not collect these intermediaries. Second, our method to screen for shared sequences based on cross-genus alignment was based on arbitrary alignment length and identity cutoffs. As expected, we detected more clusters at shorter alignment cut-offs, and we suspect that decreasing the identity threshold would also result in the identification of more and bigger clusters. Additionally, we did not consider sequences found in different bacteria within a single genus for the purposes of cluster identification. The cross-genus parameter we employed may have also artificially enriched the number of MGEs identified among *Enterobacteriaceae*, which are known to readily undergo HGT with one another (*Cerqueira et al., 2017*). Third, we assigned MGE presence relative to single reference sequences, and based our analysis on reference sequence coverage; subsequent MGEs that either gained additional sequence or rearranged their contents would still be assigned the same MGE, even though they may have diverged substantially from the reference MGE (*Sheppard et al., 2016*). Finally, this study was based exclusively on comparative genome analyses, and the MGEs we resolved from clinical isolate genomes were not tested for their capacity to undergo HGT in vitro.

In conclusion, we have shown how bacterial whole genome sequence data, which is increasingly being generated in clinical settings, can be leveraged to study the dynamics of HGT between drug-resistant bacterial pathogens within a single hospital. Our future work will include further characterization of the shared sequences and MGEs we resolved, assessment of sequence sharing across closer genetic distances (such as within-genus transfer), exploration of MGE and host co-evolution, and incorporation of additional epidemiologic information to identify shared exposures and possible routes for MGE transfer independent from bacterial transmission. Ultimately, we aim to develop this analysis into a reliable method that can generate actionable information and enhance traditional approaches to prevent and control multidrug-resistant bacterial infections.

## Materials and methods

### Isolate collection and patient demographics

Isolates were collected through the Enhanced Detection System for Hospital-Acquired Transmission (EDS-HAT) project at the University of Pittsburgh (*Sundermann et al., 2019a*). Eligibility of bacterial isolates for genome sequencing under EDS-HAT required positive clinical culture for high-priority and multidrug-resistant bacterial pathogens with either of the following criteria: >3 hospital days after admission, and/or any procedure or prior inpatient stay in the 30 days prior to isolate collection. Bacterial isolates were collected between November 2016 and May 2018. Pathogens collected included: *Acinetobacter* spp., *Burkholderia* spp., *Citrobacter* spp., *Clostridioides difficile*, vancomycin-resistant *Enterococcus* spp., extended-spectrum beta-lactamase (ESBL)-producing *E. coli*, ESBL-producing *Klebsiella* spp., *Proteus* spp., *Providencia* spp., *Pseudomonas* spp., *Serratia* spp., *Stenotrophomonas* spp., and methicillin-resistant *S. aureus*. Eligible isolates were identified using TheraDoc software (Version 4.6, Premier, Inc, Charlotte, NC). The EDS-HAT project involves no contact with human subjects; the project was approved by the University of Pittsburgh Institutional Review Board and was classified as being exempt from informed consent.

To assess patient demographics and co-morbidities, information was collected from available patient records and was summarized by an honest broker. In order to define the severity of illness and morbidity for patients included in the study, the Charlson Comorbidity Index score was calculated using ICD-9 and ICD-10 visit diagnoses from inpatient and outpatient encounters in the one year prior to each patient's admission, including the admission during which a study isolate was collected (*Quan et al., 2005*). For patients that had multiple isolates, demographic and clinical information was reported from the date of the first isolate collected. Differences in demographic and clinical factors between patient groups were assessed using Fisher's Exact test for categorical variables and Wilcoxon rank-sum test for continuous variables.

### Whole genome sequencing and analysis

Genomic DNA was extracted from pure overnight cultures of single bacterial colonies using a Qiagen DNeasy Tissue Kit according to manufacturer's instructions (Qiagen, Germantown, MD). Illumina library construction and sequencing were conducted using the Illumina Nextera DNA Sample Prep Kit with 150 bp paired-end reads, and libraries were sequenced on the NextSeq sequencing platform (Illumina, San Diego, CA). Selected isolates were also sequenced with long-read technology on a MinION device (Oxford Nanopore Technologies, Oxford, United Kingdom). Long-read sequencing libraries were prepared and multiplexed using a rapid multiplex barcoding kit (catalog SQK-RBK004) and were sequenced on R9.4.1 flow cells. Base-calling on raw reads was performed using Albacore v2.3.3 or Guppy v2.3.1 (Oxford Nanopore Technologies, Oxford, UK).

Illumina sequencing data were processed with Trim Galore v0.6.1 to remove sequencing adaptors, low-quality bases, and poor-quality reads. Bacterial species were assigned by k-mer clustering with Kraken v1.0 (*Wood and Salzberg, 2014*) and RefSeq (*Pruitt et al., 2007*) databases. Genomes were assembled with SPAdes v3.11 (*Bankevich et al., 2012*), and assembly quality was verified using QUAST (*Gurevich et al., 2013*). All genomes generated by the EDS-HAT project during the 18-month time period from November, 2016 through May, 2018 were included in this study, as long as the genome assemblies had: (a) coverage (read depth)>40X, (b) genome length within 20% of the expected size for the genus (c) a total number of contigs less than 400 and, (d) an N50 greater than 50 kb. Genomes were annotated with Prokka v1.13 (*Seemann, 2014*). Multi-locus sequence types (STs) were assigned using PubMLST typing schemes with mlst v2.16.1 (*Jolley and Maiden, 2010*; *Seemann, 2014*), and ribosomal sequence types (rMLSTs) for isolates not assigned an ST were approximated by alignment to rMLST reference sequences. Long-read sequence data was combined with Illumina data for the same isolate, and hybrid assembly was conducted using Unicycler v0.4.7 or v0.4.8-beta (*Wick et al., 2017*).

### Identification and phylogenetic analysis of shared sequence clusters

Illumina genome assemblies were screened all-by-all against one another using nucmer v4.0.0beta2 (*Marçais et al., 2018*). The nucmer output was filtered to only include alignments between isolates of different bacterial genera of at least 5,000 bp at 100% identity. Nucleotide sequences from the

resulting alignments were then extracted and compared against one another by all-by-all BLASTn v2.7.1 (*Altschul et al., 1990*). Results were filtered to only include nucleotide sequences having 100% identity over at least 5000 bp to at least one sequence from another genus. The resulting comparisons were clustered and visualized using Cytoscape v3.7.1 (*Shannon et al., 2003*). A phylogeny of shared sequence cluster-encoding genomes was constructed using the Genome Taxonomy Database Tool Kit (GTDBTK) (*Parks et al., 2018*). Briefly, translated amino acid sequences of 120 ubiquitous bacterial genes were generated, concatenated, and aligned using GTDBTK's *identify* pipeline. The resulting multiple sequence alignment was masked for gaps and uncertainties, then a phylogenetic tree was generated using RAxML v8.0.26 with the PROTGAMMA substitution model (*Stamatakis, 2014*) and 1000 iterations. Additional core genome phylogenies were generated for ST258 *K. pneumoniae* and ST131 *E. coli* genomes using snippy (v4.4.5; https://github.com/tseemann/snippy) and RAxML (*Stamatakis, 2014*).

## Characterization of shared sequences and assignment of MGEs

The longest nucleotide sequence in each shared sequence cluster was considered representative of that cluster, and was annotated with Prokka v1.13. Representative sequences were compared to publicly available genomes by BLASTn v2.7.1 against the NCBI Nucleotide database. Antibiotic resistance genes were identified by a BLASTn-based search against the CARD v3.0.1 (*Jia et al., 2017*) and ResFinder v3.2 (*Zankari et al., 2012*) databases. Plasmid replicons and MOB families were identified by a BLASTn-based search against the PlasmidFinder database v2.0.2 (*Carattoli et al., 2014*), the plasmid MLST website (https://pubmlst.org/plasmid; *Jolley et al., 2018*), and MOBscan (*Garcillán-Barcia et al., 2020*). Additional features of each shared sequence cluster were identified by consulting annotations assigned by Prokka. Sequences were aligned to one another using Geneious v11.1.5 (Biomatters Ltd., Auckland, New Zealand) and EasyFig v2.2.2 (*Sullivan et al., 2011*), and circular plots were generated with Circos (*Krzywinski et al., 2009*).

To resolve the MGEs encoding shared sequence clusters C1-C5, we first selected the earliest isolate containing each cluster for long-read sequencing and hybrid assembly. The closed, cluster-encoding mobile element (plasmid or chromosomal) from this earliest isolate was used as a reference for mapping contigs from Illumina assemblies from all other isolates using BLASTn. Briefly, contigs from Illumina-only assemblies were aligned to each reference MGE, and MGEs were called present in isolates having at least 90% coverage of a reference MGE. Among isolates having less than 90% coverage, a representative was again selected for long-read sequencing and hybrid assembly, and the process was repeated until all 104 isolates had been assigned to a MGE. Names of MGEs include the MGE type (c = chromosomal, p=plasmid), the reference isolate, and the hybrid assembly contig number, denoted with an underscore at the end of the name. Plasmids resolved through hybrid assembly were also used as reference sequences to query their presence in the entire 2173 genome data set using the same BLASTn coverage-based analysis as above, using a 90% coverage cut-off based on mapping of contigs from Illumina assemblies. When isolate genomes showed high coverage of multiple reference plasmids, the longest plasmid having at least 90% coverage was recorded. For the coverage-based analysis, we considered all isolates, regardless of whether or not their MGEs were shared across genera.

## Systematic chart review to assess epidemiologic links between patients with the same plasmids

Patients whose isolates carried the two plasmids found to putatively transfer within individual patients were reviewed using a systematic approach modified from previously published methodologies examining patient locations and procedures for potential similarities (*Eyre et al., 2013*; *Ward et al., 2019*). Patients were considered infected/colonized with the recovered plasmid on the day of the patients' culture and all subsequent days. Potential transfer events were considered significant for locations if an uninfected/uncolonized patient was housed on the same unit location or service line location (units with shared staff) at the same time or different time as a patient infected/colonized with the plasmid, using a 60-day window prior to the newly infected/colonized patient's culture date. Additionally, procedures (e.g. operating room procedures, bedside invasive procedures) were evaluated for commonalities among all patients 60 days prior to infection/colonization, as well as potential procedures contaminated by prior infected/colonized patients that could have

transferred to newly infected/colonized patients, again using a 60-day window prior to the culture date. Procedures were deemed significant if >1 patient had a similar procedure, or if there was a shared procedure within the 60-day window.

## Acknowledgements

We gratefully acknowledge Chinelo Ezeonwuka, Daniel Snyder, Jieshi Chen, Hayley Nordstrom, and Alfonso Santos-Lopez for their helpful contributions to this study. This work was supported by a grant from the Competitive Medical Research Fund of the UPMC Health System to DVT, by NIAID grants R21Al109459 and R01AI127472 to LHH and U01AI124302 to VSC, and by the Department of Medicine at the University of Pittsburgh School of Medicine. The funders had no role in study design, data collection and interpretation, or the decision to submit the work for publication.

## Additional information

### Funding

| Funder | Grant reference number | Author |
| --- | --- | --- |
| University of Pittsburgh Medical Center | Competitive Medical Research Fund | Daria Van Tyne |
| National Institute of Allergy and Infectious Diseases | R21Al109459 | Lee H Harrison |
| University of Pittsburgh | | Daria Van Tyne |
| National Institute of Allergy and Infectious Diseases | R01AI127472 | Lee H Harrison |
| National Institute of Allergy and Infectious Diseases | U01AI124302 | Vaughn S Cooper |

The funders had no role in study design, data collection and interpretation, or the decision to submit the work for publication.

### Author contributions

Daniel R Evans, Conceptualization, Resources, Data curation, Formal analysis, Validation, Investigation, Visualization, Methodology, Writing - original draft, Writing - review and editing; Marissa P Griffith, Data curation, Software, Formal analysis, Validation, Investigation, Methodology, Writing - review and editing; Alexander J Sundermann, Formal analysis, Investigation, Writing - review and editing; Kathleen A Shutt, Melissa I Saul, Data curation, Formal analysis, Writing - review and editing; Mustapha M Mustapha, Conceptualization, Data curation, Formal analysis, Investigation, Writing - review and editing; Jane W Marsh, Conceptualization, Investigation, Project administration, Writing - review and editing; Vaughn S Cooper, Conceptualization, Investigation, Writing - review and editing; Lee H Harrison, Conceptualization, Supervision, Funding acquisition, Investigation, Project administration, Writing - review and editing; Daria Van Tyne, Conceptualization, Data curation, Formal analysis, Supervision, Funding acquisition, Investigation, Visualization, Methodology, Writing - original draft, Project administration, Writing - review and editing

### Author ORCIDs

Vaughn S Cooper (iD) http://orcid.org/0000-0001-7726-0765
Daria Van Tyne (iD) https://orcid.org/0000-0001-7284-0103

### Ethics

Human subjects: Isolates were collected through the Enhanced Detection System for Hospital-Acquired Transmission (EDS-HAT) project at the University of Pittsburgh. The EDS-HAT project involves no contact with human subjects; the project was approved by the University of Pittsburgh Institutional Review Board and was classified as being exempt from informed consent. De-identified patient IDs and culture dates were utilized in downstream analysis.

Decision letter and Author response
Decision letter https://doi.org/10.7554/eLife.53886.sa1
Author response https://doi.org/10.7554/eLife.53886.sa2

# Additional files

## Supplementary files

• Supplementary file 1. List of isolates and accession information for Illumina sequence data.

• Supplementary file 2. Genus distribution and gene content of shared sequence clusters.

• Supplementary file 3. Accession information for hybrid assembled genomes.

• Supplementary file 4. Plasmid coverage among all isolates with >90% coverage of at least one plasmid.

• Transparent reporting form

## Data availability

Bacterial genome sequencing data have been deposited to relevant NCBI databases (SRA/GenBank).

The following dataset was generated:

| Author(s) | Year | Dataset title | Dataset URL | Database and Identifier |
|---|---|---|---|---|
| van Tyne D | 2020 | Systematic analysis of cross-genus horizontal gene transfer among bacterial pathogens in a single hospital | https://www.ncbi.nlm.nih.gov/bioproject/?term=PRJNA609916 | NCBI BioProject, PRJNA609916 |

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
