## [Decision Letter]

**Acceptance summary:**

This paper shows the process of horizontal gene transfer within a hospital at an unprecedented level of detail and resolution.

**Decision letter after peer review:**

Thank you for submitting your article "Comprehensive analysis of horizontal gene transfer among multidrug-resistant bacterial pathogens in a single hospital" for consideration by *eLife*. Your article has been reviewed by three peer reviewers, including Marc Lipsitch as the Reviewing Editor and Reviewer #1, and the evaluation has been overseen by Neil Ferguson as the Senior Editor.

The reviewers have discussed the reviews with one another and the Reviewing Editor has drafted this decision to help you prepare a revised submission.

Summary:

This paper describes an extensive sequencing project of over 2000 bacterial isolates including the mobile elements from an 18-month period at the University of Pittsburgh hospital. Specifically, authors screened 2173 genomes by using an all-by-all alignment with nucmer to identify shared regions of >5kb and 100% identity. Shared sequences were found in 192 genomes across 11 genera, which were then grouped into 51 clusters of related sequences (ranging from 2-52 genomes in a cluster, with 2-4 genera). Within these clusters, they selected a sample for long-read sequencing, to resolve plasmids, and identified 17 plasmids, with 10 present in >1 sample. They then aligned short reads from all samples to these plasmids, to predict which were present in each. For two patients, authors identified epidemiological links suggesting potential transfer of plasmids within a species, as well as two across-species transfers within the same patients. The identification of plasmid transfer both within and between patients is clinically relevant and highlights the value of incorporating long-read sequencing data in hospital-based surveillance for infection control. Combining these sequencing results with epidemiologic data and enhancing them with hybrid assemblies to understand plasmid transfer, the authors compose a portrait of horizontal gene transfer in a well-defined hospital population that is more extensive than any of which I am aware. As a descriptive study and a data set that will be rich for further analysis, this is a remarkable piece of work which can be made better with some further analysis and improved presentation.

The revisions below are taken from two of the three reviews, which found the manuscript valuable but in need of some revision. For the authors' information, we note a dissenting view from one reviewer, who found the significance of the paper to be less. They wrote: "Notwithstanding the scope of the sampling and numbers of strains examined, there is not much in the way of new information or novel findings. The dissemination of plasmids, mobile elements, antibiotic resistance gene, etc. in hospitals and other settings is a topic of numerous publications, and the degree to which genes can be transferred within and among species, sometimes between distantly related species, is well established." They further noted the uncertainty of the transmission inferences and questioned how generalizable the study would be to other kinds of institutions. These points are noted but the opinion of the reviewing editor is that notwithstanding these caveats, the scale and completeness of the study give it adequate importance to be publishable in *eLife*, pending revisions.

Essential revisions:

1) At no point in the paper is any justification given for the choices made about the limitation to MDR isolates (or really what that means), the 5000 bp identity requirement, the exclusive focus on inter-genera transfer (while the paper does in fact offer some tantalizing discussion of within-genus transfer), or the very limited information given about clusters after the first 5. Also, the Materials and methods do not describe how the decision to further investigate e.g. epidemiologic links or extent of sequence overlap (pairwise versus fully connected) was made. This manuscript is well-organized but gives a bit of an impression that the sequencing was done, and then some lines of inquiry that seemed inquiry were followed up, until a certain amount of time/effort/results, and a paper was written. This is a perfectly fine thing to do, especially with such interesting material, but it leaves the reader wondering what the full story is. The title says, "comprehensive analysis…among multi-drug resistant pathogens in a single hospital." There is a lot of bioinformatic analysis, but the phrase "comprehensive analysis" would suggest that more things were measured and quantified, and the word comprehensive would suggest that within-genus transfers were also identified and considered. Some specific questions that should be answered if possible are:

– Was there a single definition of an inferred transfer event? Is being in a cluster necessary and sufficient for that inference? Should other definitions be considered?

– Is the inferred amount of HGT between genera high or low or intermediate? Compared to what prior estimates?

– How does the amount of HGT between genera compare to the amount inferred within a genus (perhaps normalized for opportunities to see this)?

– What proportion of inferred events have a plausible path of epidemiologic links?

– What is the impact of the 5kb identity requirement – do you get radically different answers with 3 or 10 as cut-offs?

– What descriptions can be given in quantitative terms of the different patterns of what was conserved within a cluster between C1 and C2, C3? What about all the other clusters?

– What proportion of the inferred events involve mechanisms of horizontal transfer consistent with what we already know (e.g. plasmid transfer, ICE, etc), and which are unexplained by those?

– How do the inferred rates of evolution from the inferred transfer events compare with what we know? The Tn7 comment is tantalizing but is one example – a comprehensive analysis would consider the overall patterns.

– What was the extent of movement of MDR determinants together among inferred events?

Perhaps not all of these can be answered, but to ignore them seems unfortunate in a paper with such a grand title. I don't want to dictate the publication strategy for what will undoubtedly be a series of papers, but a paper in *eLife* that is called a "Comprehensive Analysis" should not, for example, consider only transfer between genera.

2) Authors state the epidemiology of MGEs in clinical settings requires detailed individual level data, but actually provide nearly no epidemiological data in the current manuscript. I would have expected a table at a minimum outlining the demographics and clinical characteristics of the patients included (N=2173). It would also be helpful to know more about the demographics and clinical characteristics of the patients whose isolates share sequences by cluster (N-192). For example, looking at Figure 1B, I note there are 13 clusters containing *Stenotrophomonas* – 12 of which are clustered only with *Pseudomonas*. This would suggest to me that these may be patients with Cystic Fibrosis, as both pathogens are commonly found in the CF lung, but it would be helpful to know this information to better assess the clinical relevance of this work.

3) Authors did long read sequencing on a subset of "representative isolates from the largest clusters" – what do authors mean by “representative” here? Do they just mean they chose a random sample from within each cluster? Please explain.

4) Much of the interpretive material is confusing or questionable.

– The sentence “Taken together, these results indicate that while many of the sequences we identified were shared between related bacterial genera, our approach also identified sequences that were identical in the genomes of distantly related pathogens.” took about four reads before I decided it simply meant there was a lot of sharing among close genera, and some sharing among more distant genera. A more parallel structure to the sentence could clarify (if that is indeed what it means). A reference to Figure 2 could also help.

– Discussion section: "generate biased interpretations of the driving forces": I don't see any interpretation of the driving forces behind HGT (or as I imagine driving forces, behind the success of lineages which have undergone HGT, such as transmissibility or antimicrobial selection pressure or the like) in this paper, and moreover, biased interpretations can be biased only relative to some defined estimand. This seems like unduly vague language, and maybe should be replaced with "incomplete accounts of the extent of HGT" or something similar.

– Discussion paragraph three is somewhat peculiar. It seems to rest on the premise that if an element moves to a new host, it will be selected to change its sequence to adapt to that host, but then undermines that premise with hypotheses 2 and 3. This may be just a matter of wording, but it seems confusing. Maybe sequences are adapted to generic functions (e.g. neutralizing a drug) rather than to the bacterial host. At a minimum the wording should be changed; better would be, instead of giving one example of each, to try to classify the clusters based on these explanations. Again, this is part of the distance between "comprehensive analysis" and the more descriptive tone of the paper.

– "both plasmids…from the same patient were nearly identical to one another, suggesting that they were indeed transferred shortly before the bacteria were isolated" – over what timescale might we expect difference to occur in plasmids and of what magnitude?

– "underscores how quickly MGEs can move" – this makes no sense. MGEs move by for example conjugation which has been measured in the lab as taking minutes. The literal movement is of course fast. I think finding evidence of transfer that is close together in space and time is unremarkable; finding persistence over time is more remarkable.

– Some of the other conclusions seem a bit unsupported by the analyses that are conducted – e.g. "the fact we only observed plasmids in closely related bacterial lineages suggests that they are well-adapted to these lineages, and if they were transmitted to other STs they were likely lost due to instability and/or fitness costs". I would think this could easily be affected by sampling strategy used, with only invasive samples of select species.

– Figure 1D and E just show the proportion of clusters with X gene type or Y AME gene, respectively, but underlying cluster sizes range from 2-52 – shared across all samples. Is there a way authors can standardize by cluster size, as I am not sure a cluster of 2 should have the same weight as a cluster of 52 in these analyses?

– Authors required a minimum of 5kb and 100% identity on nucmer and state these cut-offs were arbitrary (Discussion) – did authors examine any other cut-offs and how do their findings change if these are adjusted?

5) Authors report that they aligned short-reads from all of the isolates to the reference sequences they generated for cluster-containing MGEs (chromosomal or plasmid). They then assessed the coverage (whether this is the% alignment to the reference or depth is unclear – please clarify) to these references to predict whether the respective genome contained the MGE or not. However, I could not find a table showing these results. It would be very helpful to have, for each isolate, the percent of each of these references covered and the median depth of coverage in order to assess the reliability of these results. At a minimum, this should be provided for the plasmid analysis, wherein they found 93 isolates had cluster C1-C5 sequences of 17 plasmids.

6) Transmission analyses – Given the high percent identity between plasmids, the results suggesting transmission of plasmids between patient A and B look quite convincing. However, it would help to have the dates of sample collection for each of these samples. Currently, authors just state patient B's isolate was collected after patient A's. As these are invasive isolates only, it is also is possible that transmission occurred via a colonized intermediary who was not detected due to the study design.

7) A major conclusion authors reach is that some plasmids carrying putative MGEs "were likely inherited vertically as bacteria were transmitted between patients in the hospital." I am a bit surprised not to see any analysis to assess whether the bacterial chromosomes were indeed the same as well as the plasmid given authors have complete Illumina data for these isolates. Authors could easily align reads to a chromosome reference from the respective ST and assess whether this is indeed probable, rather than speculating based on the MGE/plasmid analysis alone.

[Editors' note: further revisions were suggested prior to acceptance, as described below.]

Thank you for resubmitting your work entitled "Systematic detection of horizontal gene transfer across genera among multidrug-resistant bacteria in a single hospital" for further consideration by *eLife*. Your revised article has been evaluated by Neil Ferguson (Senior Editor) and a Reviewing Editor.

The manuscript has been improved but there are some remaining issues that need to be addressed before acceptance, as outlined below:

Specifically, we have one remaining concern regarding the 5kb threshold for the cluster analysis. The 5kb threshold (which was used in your original analysis), produced 51 clusters. As requested by reviewers, in the revised manuscript the clustering results with >3kb and >10kb are now shown. However, compared to the 5kb threshold, using a 3kb threshold results in 120 clusters, while using a 10kb threshold results in only 16. Despite finding such a wide range in clustering due to this, this has not been discussed anywhere in the current paper – or the implications this would have on the conclusions. It therefore seems like the results are very sensitive to the threshold used, so this should at least be discussed in the paper as a limitation of this work.

---

## [Author Response]

Essential revisions:1) At no point in the paper is any justification given for the choices made about the limitation to MDR isolates (or really what that means), the 5000 bp identity requirement, the exclusive focus on inter-genera transfer (while the paper does in fact offer some tantalizing discussion of within-genus transfer), or the very limited information given about clusters after the first 5. Also, the Materials and methods do not describe how the decision to further investigate e.g. epidemiologic links or extent of sequence overlap (pairwise versus fully connected) was made. This manuscript is well-organized but gives a bit of an impression that the sequencing was done, and then some lines of inquiry that seemed inquiry were followed up, until a certain amount of time/effort/results, and a paper was written. This is a perfectly fine thing to do, especially with such interesting material, but it leaves the reader wondering what the full story is. The title says, "comprehensive analysis…among multi-drug resistant pathogens in a single hospital." There is a lot of bioinformatic analysis, but the phrase "comprehensive analysis" would suggest that more things were measured and quantified, and the word comprehensive would suggest that within-genus transfers were also identified and considered. Some specific questions that should be answered if possible are:– Was there a single definition of an inferred transfer event? Is being in a cluster necessary and sufficient for that inference? Should other definitions be considered?– Is the inferred amount of HGT between genera high or low or intermediate? Compared to what prior estimates?– How does the amount of HGT between genera compare to the amount inferred within a genus (perhaps normalized for opportunities to see this)?– What proportion of inferred events have a plausible path of epidemiologic links?– What is the impact of the 5kb identity requirement – do you get radically different answers with 3 or 10 as cut-offs?– What descriptions can be given in quantitative terms of the different patterns of what was conserved within a cluster between C1 and C2, C3? What about all the other clusters?– What proportion of the inferred events involve mechanisms of horizontal transfer consistent with what we already know (e.g. plasmid transfer, ICE, etc), and which are unexplained by those?– How do the inferred rates of evolution from the inferred transfer events compare with what we know? The Tn7 comment is tantalizing but is one example – a comprehensive analysis would consider the overall patterns.– What was the extent of movement of MDR determinants together among inferred events?Perhaps not all of these can be answered, but to ignore them seems unfortunate in a paper with such a grand title. I don't want to dictate the publication strategy for what will undoubtedly be a series of papers, but a paper in eLife that is called a "Comprehensive Analysis" should not, for example, consider only transfer between genera.

We appreciate this feedback and agree that the title of our original submission was perhaps misleading. We have modified the title to more accurately reflect what we have actually done in this study. We have added a definition of inferred transfer events and have incorporated a discussion of how little is currently known about plasmid transfer rates in vivo in the Results section. To address these comments more generally, we have deposited all of the Illumina sequence data, as well as the hybrid assembled genomes we generated, into relevant NCBI databases (see Supplementary files 1 and 3). In making the sequence data from our study publicly available, anyone who wishes to will be able to conduct their own analyses (using whatever parameters they choose) on this dataset. Nonetheless, we have repeated our cluster analysis using 3kb and 10kb cut-offs and have added the resulting shared sequence clusters as an additional figure (Figure 1—figure supplement 1). To address the reviewer’s comments about cluster contents and possible mechanisms of transfer, we have added a table describing the gene content of each cluster to the revised manuscript (Supplementary file 2). As we state in the Discussion, we plan to further investigate within-genus transfer in future work. With all of our sequence data publicly available, however, other researchers are free to query this dataset in whatever way(s) they might like.

2) Authors state the epidemiology of MGEs in clinical settings requires detailed individual level data, but actually provide nearly no epidemiological data in the current manuscript. I would have expected a table at a minimum outlining the demographics and clinical characteristics of the patients included (N=2173). It would also be helpful to know more about the demographics and clinical characteristics of the patients whose isolates share sequences by cluster (N-192). For example, looking at Figure 1B, I note there are 13 clusters containing Stenotrophomonas – 12 of which are clustered only with Pseudomonas. This would suggest to me that these may be patients with Cystic Fibrosis, as both pathogens are commonly found in the CF lung, but it would be helpful to know this information to better assess the clinical relevance of this work.

The is an excellent point. We have now included a table that summarizes relevant demographic and clinical information from the patients in the study (Table 1).

3) Authors did long read sequencing on a subset of "representative isolates from the largest clusters" – what do authors mean by “representative” here? Do they just mean they chose a random sample from within each cluster? Please explain.

We selected isolates for long-read sequencing by considering the length of the shared sequence (i.e. the isolate with the longest sequence was selected), as well as culture date (i.e. selecting the earliest isolate). We have included this information in the revised manuscript Results section.

4) Much of the interpretive material is confusing or questionable.– The sentence “Taken together, these results indicate that while many of the sequences we identified were shared between related bacterial genera, our approach also identified sequences that were identical in the genomes of distantly related pathogens.” took about four reads before I decided it simply meant there was a lot of sharing among close genera, and some sharing among more distant genera. A more parallel structure to the sentence could clarify (if that is indeed what it means). A reference to Figure 2 could also help.

We apologize for the confusion. This sentence has been removed from the revised manuscript.

– Discussion section: "generate biased interpretations of the driving forces": I don't see any interpretation of the driving forces behind HGT (or as I imagine driving forces, behind the success of lineages which have undergone HGT, such as transmissibility or antimicrobial selection pressure or the like) in this paper, and moreover, biased interpretations can be biased only relative to some defined estimand. This seems like unduly vague language, and maybe should be replaced with "incomplete accounts of the extent of HGT" or something similar.

We agree and have modified this sentence accordingly.

– Discussion paragraph three is somewhat peculiar. It seems to rest on the premise that if an element moves to a new host, it will be selected to change its sequence to adapt to that host, but then undermines that premise with hypotheses 2 and 3. This may be just a matter of wording, but it seems confusing. Maybe sequences are adapted to generic functions (e.g. neutralizing a drug) rather than to the bacterial host. At a minimum the wording should be changed; better would be, instead of giving one example of each, to try to classify the clusters based on these explanations. Again, this is part of the distance between "comprehensive analysis" and the more descriptive tone of the paper.

We thank the reviewer for this helpful feedback. The three possible causes that we listed are hypotheses that might help explain our observations, and for which we have some (very limited) evidence in our dataset. We chose to offer some speculation regarding what we feel is the strongest example supporting each hypothesis; formally testing each one, especially in a dataset of this size, would require significant additional work that we feel would be better suited to future studies. We have nonetheless modified this section of the manuscript accordingly.

– "both plasmids…from the same patient were nearly identical to one another, suggesting that they were indeed transferred shortly before the bacteria were isolated" – over what timescale might we expect difference to occur in plasmids and of what magnitude?

This is an important point to consider. Unfortunately, careful estimates of mutation rates among mobile elements are currently lacking. We have revised this section of the manuscript accordingly.

– "underscores how quickly MGEs can move" – this makes no sense. MGEs move by for example conjugation which has been measured in the lab as taking minutes. The literal movement is of course fast. I think finding evidence of transfer that is close together in space and time is unremarkable; finding persistence over time is more remarkable.

This is a fair point and we have modified this sentence accordingly.

– Some of the other conclusions seem a bit unsupported by the analyses that are conducted – e.g. "the fact we only observed plasmids in closely related bacterial lineages suggests that they are well-adapted to these lineages, and if they were transmitted to other STs they were likely lost due to instability and/or fitness costs". I would think this could easily be affected by sampling strategy used, with only invasive samples of select species.

This is also a fair point, and we have removed this sentence from the revised manuscript.

– Figure 1D and E just show the proportion of clusters with X gene type or Y AME gene, respectively, but underlying cluster sizes range from 2-52 – shared across all samples. Is there a way authors can standardize by cluster size, as I am not sure a cluster of 2 should have the same weight as a cluster of 52 in these analyses?

We agree that cluster size might affect how we interpret the contents of each cluster. We now include a table that summarizes the number of isolates and genetic contents of each cluster (Supplementary file 2), which will allow readers to access this information directly.

– Authors required a minimum of 5Kb and 100% identity on nucmer and state these cut-offs were arbitrary (Discussion) – did authors examine any other cut-offs and how do their findings change if these are adjusted?

In line with this and the comment above, we repeated our cluster detection using 3kb and 10kb cut-offs and have included the results in the revised manuscript in the Results section (also see Figure 1—figure supplement 1). In the future we plan to explore different identity cut-offs as well.

5) Authors report that they aligned short-reads from all of the isolates to the reference sequences they generated for cluster-containing MGEs (chromosomal or plasmid). They then assessed the coverage (whether this is the% alignment to the reference or depth is unclear – please clarify) to these references to predict whether the respective genome contained the MGE or not. However, I could not find a table showing these results. It would be very helpful to have, for each isolate, the percent of each of these references covered and the median depth of coverage in order to assess the reliability of these results. At a minimum, this should be provided for the plasmid analysis, wherein they found 93 isolates had cluster C1-C5 sequences of 17 plasmids.

We thank the reviewer for this feedback. To assess reference sequence presence, we aligned contigs from Illumina assemblies to each reference sequence, rather than raw reads. Because of this there was no cut-off used for read depth. We have included a table summarizing the coverage results from our contig-mapping analysis (Supplementary file 4), and both the raw read data for all study isolates and the reference plasmid sequences have been posted to SRA/GenBank, so that others can conduct analyses with this data.

6) Transmission analyses – Given the high percent identity between plasmids, the results suggesting transmission of plasmids between patient A and B look quite convincing. However, it would help to have the dates of sample collection for each of these samples. Currently, authors just state patient B's isolate was collected after patient A's. As these are invasive isolates only, it is also is possible that transmission occurred via a colonized intermediary who was not detected due to the study design.

We agree that providing additional information about the plasmid transferred between Patients A and B would be helpful. We have added information about the time separating the collection of their isolates in the Results section, as well as additional speculation that transfer could have happened during colonization, which we did not assess, in the Discussion section.

7) A major conclusion authors reach is that some plasmids carrying putative MGEs "were likely inherited vertically as bacteria were transmitted between patients in the hospital." I am a bit surprised not to see any analysis to assess whether the bacterial chromosomes were indeed the same as well as the plasmid given authors have complete Illumina data for these isolates. Authors could easily align reads to a chromosome reference from the respective ST and assess whether this is indeed probable, rather than speculating based on the MGE/plasmid analysis alone.

This is a valid point. We have looked more closely at the *K. pneumoniae* ST258 and *E. coli* ST131 genomes in our dataset and found that in both cases, the isolates carrying these plasmids were more closely related to each other than to other isolates lacking the plasmids. This finding supports our conclusion that the plasmids were likely transmitted along with the bacteria carrying them and has been added to the revised manuscript in the Results section (see also Figure 5—figure supplement 1).

[Editors' note: further revisions were suggested prior to acceptance, as described below.]

The manuscript has been improved but there are some remaining issues that need to be addressed before acceptance, as outlined below:Specifically, we have one remaining concern regarding the 5kb threshold for the cluster analysis. The 5kb threshold (which was used in your original analysis), produced 51 clusters. As requested by reviewers, in the revised manuscript the clustering results with >3kb and >10kb are now shown. However, compared to the 5kb threshold, using a 3kb threshold results in 120 clusters, while using a 10kb threshold results in only 16. Despite finding such a wide range in clustering due to this, this has not been discussed anywhere in the current paper – or the implications this would have on the conclusions. It therefore seems like the results are very sensitive to the threshold used, so this should at least be discussed in the paper as a limitation of this work.

This is an important point. Cluster number will also vary based on the identity threshold used – if we lowered the nucleotide identity threshold below 100% we would almost certainly detect more (and bigger) clusters at all length cut-offs. We agree that this is an additional limitation that warrants attention and have incorporated this idea into both the Results and Discussion sections in the revised manuscript.